

# Evaluating the mesoscale spatio-temporal variability in simulated wind speed time series over Northern Europe

Graziela Luzia[1], Andrea N. Hahmann[1], and Matti Juhani Koivisto[1]

[1]Department of Wind and Energy Systems, Technical University of Denmark, Roskilde, Denmark

**Correspondence:** Graziela Luzia (gldco@dtu.dk)

**Abstract.** As wind energy increases its share of total electricity generation and its integration into the power system becomes more challenging, accurately representing the spatio-temporal variability in wind data becomes crucial. Wind fluctuations impact power and energy systems, e.g., energy system planning, vulnerability to storm shutdowns, and available voltage stability support. To analyze such fluctuations and their spatio-temporal dependencies, time series of wind speeds at hourly time-

frequency or higher are needed. We provide a comprehensive evaluation of the global and mesoscale-model derived wind time series against observations by using a set of metrics that we present as requirements for wind energy integration studies. We also perform a sensitivity analysis to find the best model setup of the Weather Research and Forecasting (WRF) model, focusing on evaluating the wind speed fluctuation metrics. The results show that using higher spatial resolution in the WRF model simulations improves the representation of temporal fluctuations; however, higher-resolution simulations often lower the

correlations of wind time series with measurements. We also show that the nesting strategy is an important consideration, and a smoother transition from the forcing data to the nested domains improves the correlations with measurements. All mesoscale model simulations overestimate the value of the spatial correlations in wind speed with respect to their observed values. Still, the spatial correlations and the wind speed distributions are insensitive to the model configuration tested in this study.

## 1 Introduction

Many wind energy applications use meteorological data derived from atmospheric models; for example, in the production of wind resource atlases (Tammelin et al., 2013; Dörenkämper et al., 2020; Solbrekke et al., 2021), and extreme wind atlases (Larsén et al., 2012), and wind turbine icing in cold climates (Hämäläinen and Niemelä, 2017). In general, most studies use time series of wind speed, but different applications require distinct qualities in meteorological data and, therefore, different evaluations are necessary. For a wind resource atlas, for instance, accurate wind speed distributions are necessary (Dörenkämper

et al., 2020; Knoop et al., 2020); for wind power forecasting, accurate timing is vital (Das et al., 2017; Olson et al., 2019). Energy and power system modeling, including optimal energy system planning, (e.g., Gea-Bermúdez et al., 2020; Brown et al., 2018; Malvaldi et al., 2017), the study of power system ramp rates and vulnerability to storm shutdowns (e.g., Murcia et al., 2021) and available voltage stability support (e.g., Souxes et al., 2019), require an accurate representation of temporal dependencies and spatial correlations, and accurate wind speed distributions. Time synchronization with measurements (i.e.,





the high correlation between measured and simulated data) is important when the wind time series are used in conjunction with other data, e.g., electricity load time series (Gea-Bermúdez et al., 2020).

Several studies have validated meteorological data sets for the specific purpose of modeling weather-dependent wind power generation and its highly fluctuating behavior. These works use data provided by global atmospheric reanalysis (e.g., Cannon et al., 2015; González-Aparicio et al., 2017; Gruber et al., 2022), mesoscale numerical weather prediction (NWP) models

(Murcia Leon et al., 2021; Koivisto et al., 2021) or both (Jourdier, 2020; Murcia et al., 2022). Because mesoscale NWP models cannot represent the effects of the most detailed microscale processes, extra information can be added by combining microscale with mesoscale data (e.g., Staffell and Pfenninger, 2016; Ruiz et al., 2019). Due to its relatively low temporal resolution (usually available from 30 minutes to 1-hour resolution) and intrinsic numerical smoothing, data from mesoscale models cannot include short-term variability. Therefore, it may be necessary to combine synthetic data through statistical methods (e.g., Hawker et al.,

2016; Larsén et al., 2012; Murcia et al., 2021) to represent wind fluctuations at shorter time scales.

Validation studies of time series from existent high spatial resolution data sets (in the order of a few kilometers) produced by mesoscale NWP models can be found in the literature for wind power integration studies. Jourdier (2020) compared data sets from several sources, including large scale and regional downscaled reanalyses, with mesoscale data such as the NWP model AROME and the New European Wind Atlas (NEWA) in simulating mean wind speed, power production, and its temporal

correlations over France. Murcia et al. (2022) performed a large scale validation study, comparing the ERA5 reanalysis and two data sets based on the Weather Research and Forecasting (WRF), NEWA, and an ERA-Interim (Dee et al., 2011) based European-level atmospheric reanalysis (Nuño et al., 2018) with and without the addition of microscale details provided by the Global Wind Atlas (Badger et al., 2015). Regarding wind speed, validations were done using wind measurements over Northern Europe on various time-series metrics, such as errors in autocorrelations, spatial correlations, and wind speed distribution.

There are fewer articles presenting the impacts of modeling development for time series focused on wind integration studies. Draxl and Clifton (2015) discussed that many efforts had been made for modeling the wind distributions for wind resource assessments with NWP models while creating data sets for wind integration studies that are time-synchronized with real profiles and capable of simulating wind power variability is a not straightforward step from the NWP outputs. Some studies utilize the mesoscale model to generate time series and validate it for an individual application. For example, Nuño et al. (2018) developed

hourly time series of European transcontinental wind and solar photovoltaic generation using the WRF model to dynamically downscale a global reanalysis and analyzed regionally aggregated power variability in different time scales, which is relevant for system planning, market studies, and others. Mehrens et al. (2016) assessed the WRF model ability in simulating coherence, spatial correlation, and power spectra in a large range of distances over the North and Baltic Sea. Focusing on local wind power generation variability, Koivisto et al. (2021) validated the WRF model time series aggregated for several countries in Europe.

The results show that combining mesoscale and microscale data and the addition of missing power plant technical parameters through machine learning improve the representation of the annual capacity factors and hourly generation distributions for most countries. However, no significant differences are shown for the auto and spatial correlations. Draxl and Clifton (2015) generated a sub-hourly high spatial resolution data set a target for application in wind integration studies over the United States but besides the high temporal resolution, they focused the validations only on intra-day and seasonal variability, in addition





to wind speed distribution. By focusing on several essential aspects of time series, this paper adds to the development of time series modeling of wind speeds for wind integration studies and evaluation techniques. The comparison results between the models allow users to select the most appropriate modeling and data sets for different applications.

This work focus on modeling wind speed time series suitable for power and energy system applications and adds to the literature by 1) investigating the impact of the interaction between mesoscale model and its forcing data on the quality of

the resultant time series and, 2) providing a comprehensive evaluation of the different data sets, with focus on how well they can represent temporal and spatial correlations. We perform a sensitivity study of the WRF model in multiple configurations, varying the influence of the forcing global reanalysis in the simulations to understand its role and distinguish the model configuration that outperforms various time series aspects. The results are also compared to ERA5 and NEWA mesoscale data. We hypothesize that these modeling aspects, defined by the nesting choice, size, and position of the domains, impact the

accuracy of the time series more than the horizontal resolution of the model simulations.

This paper is structured as follows: Section 2 describes the simulations, the measured data, and the metrics used to generate and validate the time series; Section 3 presents the results of the time series comparisons; Section 4 presents the discussion and experiment's ranking and Section 5, the conclusions and perspectives.

## 2 Data and methods

### 2.1 Simulated data

The simulations used in this work were produced by the WRF mesoscale model versions 4.2.1 (Skamarock et al., 2019) and 3.8.1 (Skamarock et al., 2008), using the configuration for the model physics and model dynamics as in the New European Wind Atlas (NEWA) (Dörenkämper et al., 2020). All simulations use nudging to the forcing reanalysis in the outer domain as outlined in Hahmann et al. (2015). The WRF model has been broadly validated and used for wind resource assessment and

sensitivity studies for Northern Europe (Hahmann et al., 2015, 2020b; Li et al., 2021).

The purpose of the nesting experiments (hereafter named "10km", "6km", "5km" and "3.3km") is to vary the influence of the forcing data in the innermost domain by changing the grid arrangement of the simulation or to verify the impact of the grid spacing, for similar arrangements. The WRF model domains used in the simulations are presented in Fig. 1 using two nesting approaches: 1) using one nested domain (blue) and 2) using two nested domains (orange). The simulations were configured to

have the innermost domains covering the same geographical area. The parent domain (d1) has the same dimensions and position for both arrangements. However, for nested domains, the number of grid points must be proportional to the parent grid ratio (Skamarock et al., 2008), which means that even for similar arrangements (i. e., single nest with ratio 1/3 or 1/5), the innermost domains have slightly (a few km) different geographical extension. These differences were neglected for the comparisons, as it is shown in Fig. 1. The horizontal resolution of the innermost domain is a result of the nesting ratio and the resolution

jump used, as it can be seen in Table 1. We included two additional experiments to test the impact from different forcing data and WRF model versions. The forcing data provides initial and boundary conditions to the simulations. All experiments were forced by the ERA5 reanalysis (Hersbach et al., 2020) as in the NEWA configuration, except experiment "10km_erai", which



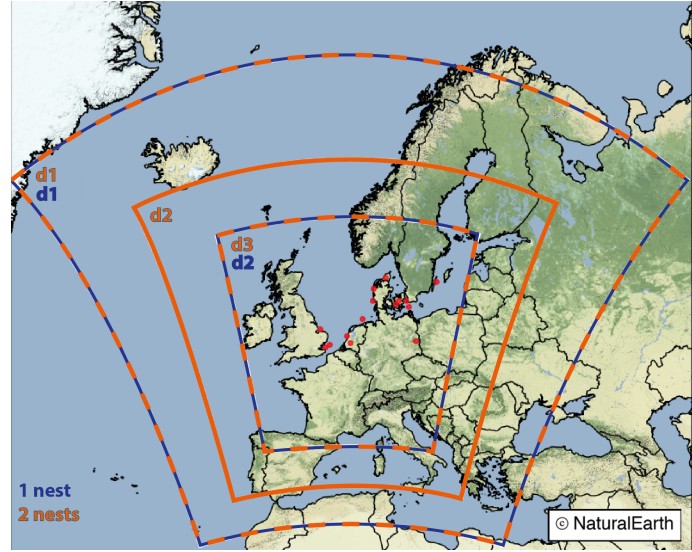

**Figure 1.** Location of the domains used in the WRF model simulations for two configurations: 1) single nest (domains 1 and 2, in blue) and 2) two nests (domains 1, 2 and 3, in orange). The wind speed measurements locations are shown by the red dots. Base map created with Natural Earth.

was forced by the ERA-Interim reanalysis (Dee et al., 2011). All experiments were repeated using WRF version 3.8.1 (as in the NEWA production) for checking the consistency among the results, but only the analogous to "10km" from version 4.2.1

(named "10km_v3") is presented here. All experiments are comparable to the "10km" with only one aspect modified, either in the nesting arrangement, resolution jump, forcing data, or model version.

Time series from NEWA and ERA5 reanalyses were included in the comparisons. All time series (from the simulations and the existent reanalyses) were extracted using a horizontal linear interpolation and a logarithmic vertical interpolation for each measurement location and its respective height.

**Table 1.** Experiment names and WRF model configuration. Nesting ratio refers to the grid of the relative parent domain as shown in Fig. 1. Resolution jump is the grid spacing of the outer and inner grids.

| Experiment | Nesting arrangement (Fig.1) | Nesting ratio | Resolution jump [km] | Forcing data | WRF version |
|---|---|---|---|---|---|
| 10km | single nest | 1/3 | 30/10 | ERA5 | 4.2.1 |
| 6km | single nest | 1/5 | 30/6 | ERA5 | 4.2.1 |
| 5km | single nest | 1/3 | 15/5 | ERA5 | 4.2.1 |
| 3.3km | two nests | 1/3/3 | 30/10/3.3 | ERA5 | 4.2.1 |
| 10km_erai | single nest | 1/3 | 30/10 | ERA-Interim | 4.2.1 |
| 10km_v3 | single nest | 1/3 | 30/10 | ERA5 | 3.8 |



## 2.2 Measured data

Data from 14 met masts over Northern Europe (Fig. 1) were processed and filtered using an adapted version of the quality control routine described in Ramon et al. (2020). The filter eliminates suspicious data or sequences of data, including implausible values or too extreme variations, freezing, or stuck instrument readings. When necessary, a rough attempt to minimize the effect of flow distortion caused by the mast on the wind speed measurements was made. When wind speed measurements are available from only one boom at one height on the mast, winds originating at $\pm 10°$ of the boom direction are filtered. At other sites, where wind speeds at one height are measured with more than one boom direction, the wind speed measurements are combined according to the wind direction to minimize flow distortion. Wind speed time series measured at heights ranging from 30 m to 140 m and originally at 10 minutes averaged resolution, were aggregated to hourly resolution. To ensure that all measured data has been reported at UTC times (information not always provided by the data source), an inspection of the cross-correlations between measured data and reanalysis (Fig. 2.f) has been done for checking suspicious shifted lags. Missing or invalid data identified in the measured time series were marked NaN also in the simulated data. The completeness of the series is shown in Table 2. The series covers one year of observations, and the year 2009 was chosen for being the period with time series from the maximum number of sites available. Anonymized stations were named according to the location: Central North Sea (CNS), South-North Sea (SNS), and Western Baltic Sea (WBS). Other information on the type of location (land, forest, coastal or offshore) and measurement device (met mast or lidar) are presented in Table 2.

**Table 2.** Observational data sets. Type: meteorological masts (M), LIDAR (L); location: coastal (C); land (L); offshore (S); forest (F). Availability [%] refers to the valid data within time coverage during 2009 after the quality control and minimization of flow distortion.

| Site | Height [m] | Availability [%] | Type | Location | Data sources |
|---|---|---|---|---|---|
| Børglum | 31.5 | 99.0 | M | C | DTU Database |
| Cabauw | 140 | 99.3 | M | L | CESAR Database |
| CNS1 | 108 | 94.9 | L | S | Hasager et al. (2013) |
| DockingShoal | 90 | 99.2 | M | C | Marine Data Exchange |
| FINO1 | 90.3 | 94.3 | M | S | FINO Offshore |
| FINO2 | 92.4 | 92.9 | M | S | FINO Offshore |
| Høvsøre | 100 | 97.4 | M | C | DTU Database |
| Lillgrund | 65 | 99.3 | M | C | DTU Database |
| Lindenberg | 98 | 99.4 | M | F | Ramon et al. (2020) |
| SNS1 | 116 | 85.9 | M | S | Hasager et al. (2013) |
| SNS2 | 72.5 | 99.4 | M | S | Hasager et al. (2013) |
| Sorø | 43* | 82.6 | M | F | DTU Database |
| Tystofte | 39 | 98.7 | M | L | DTU Database |
| WBS1 | 50 | 62.9 | M | C | commercial site |

* includes a displacement height of 20.5 m based on Dellwik et al. (2006).



## 2.3 Evaluation metrics

Different qualities in wind speed time series are required for applications in power and energy system studies, as described in Sect. 1. We analyze five aspects of data quality using error metrics defined in Murcia et al. (2022). The comparisons against observations include the six WRF model experiments, NEWA, and ERA5 data sets. The time series evaluation metrics include:

1. Correlation to measurements. Given an observed time series, $X(t,i)$, and a simulated time series, $Y(t,i)$, at time $t$ and location $i$, the Pearson correlation coefficient $\rho(X(t,i),Y(t,i))$ is calculated for each simulation with respect to the measured data (Fig. 2.e). This metric is important when the wind speed data (or the resulting generation data) needs to be correctly correlated with other time-stamped data sets (e.g., electricity price).

2. Error in the autocorrelation function. The autocorrelation functions (ACF), $\rho(X(t,i),X(t-\Delta t,i))$ and $\rho(Y(t,i),Y(t-$
$\Delta t,i))$, at location $i$, are calculated for the observed and simulated time series, respectively (Fig. 2.d). The error in the metric is computed as $\text{ACF}_X^{lag=\Delta t}$ minus $\text{ACF}_Y^{lag=\Delta t}$, where $\Delta t$ is one hour. The ACF metric represents how well the simulated data can represent temporal variability seen in the measured data.

3. Error in wind speed distribution. This metric quantifies the difference between simulated and observed wind speed distribution (Fig. 2.b). It is computed by using the Earth mover's distance (EMD) technique, introduced in Hahmann et al.
(2020b). The EMD is defined by the area between two cumulative density functions (Fig. 2.c) and, therefore, is always positive. An accurate wind speed distribution is essential for estimating a site's potential annual energy production, with a good fit also at the higher percentiles important for understanding storm shutdown risks.

4. Error in the standard deviation of the first difference. The standard deviation of the first difference time series $\sigma(X(t,i)-$
$X(t-\Delta t,i))$ and $\sigma(Y(t,i)-Y(t-\Delta t,i))$ at location $i$ are calculated for the observed and simulated time series, re-
spectively. The error is computed as the difference between the simulated and observed standard deviations. This metric describes how well the simulated data can represent the 1-hour ramps seen in the measured data.

5. Error in the spatial correlation among measurements. The correlations $\rho(X(t,i),X(t,j))$ and $\rho(Y(t,i),Y(t,j))$ are calculated for the observed and simulated time series for all pairs of sites $i,j$. The metric is computed by fitting equation $\rho_{ij}=\exp(-d_{ij}/L)$, with $d_{ij}$ and $L$ in km, to the correlations and distances (d) between the locations for both the
measured and simulated data (Fig. 2.g), and taking the ratio of the characteristic length scales, $L_Y/L_X$. The smaller the length scale, the faster the correlations decay to zero as distance increases. Modeling spatial correlations well is relevant for system integration studies, as the probability of wind speeds being low or high or ramping up or down simultaneously in multiple locations impacts the aggregate wind generation variability in the system.

Figure 2 illustrates all the metrics used in this work, but including only three data sets for simplicity. Figure 2a shows an
example of a time series for Børglum, a short mast at a coastal location in Denmark, with observed (OBS, in black), ERA5 (yellow), and NEWA (blue) data sets. Figure 2b shows the wind speed distribution for the three data sets and the computed EMD metrics for ERA5 and NEWA in respect to OBS, resulting from the area between two CDFs, as illustrated in Fig.2c.





Figure 2d shows the ACF for the first 24 hours, although only the ACF at lag = 1 h is used in the comparisons. Figure 2e illustrates the correlations among simulated and observed data. Figure 2f shows the cross-correlation function of simulated data with the observations, which was used for checking the correctness of the timestamps in every observed time series. Lastly, Fig. 2g illustrates the comparison between parameter $L$ computed for observed and simulated data from the spatial correlations over a distance of all 14 pairs of locations. Each metric described is computed for every location and experiment, as well as the median over different types of site (onshore, coastal and offshore) and the median of all locations for each metric. The final rank among all data sets for each metric is based on the medians of all sites.

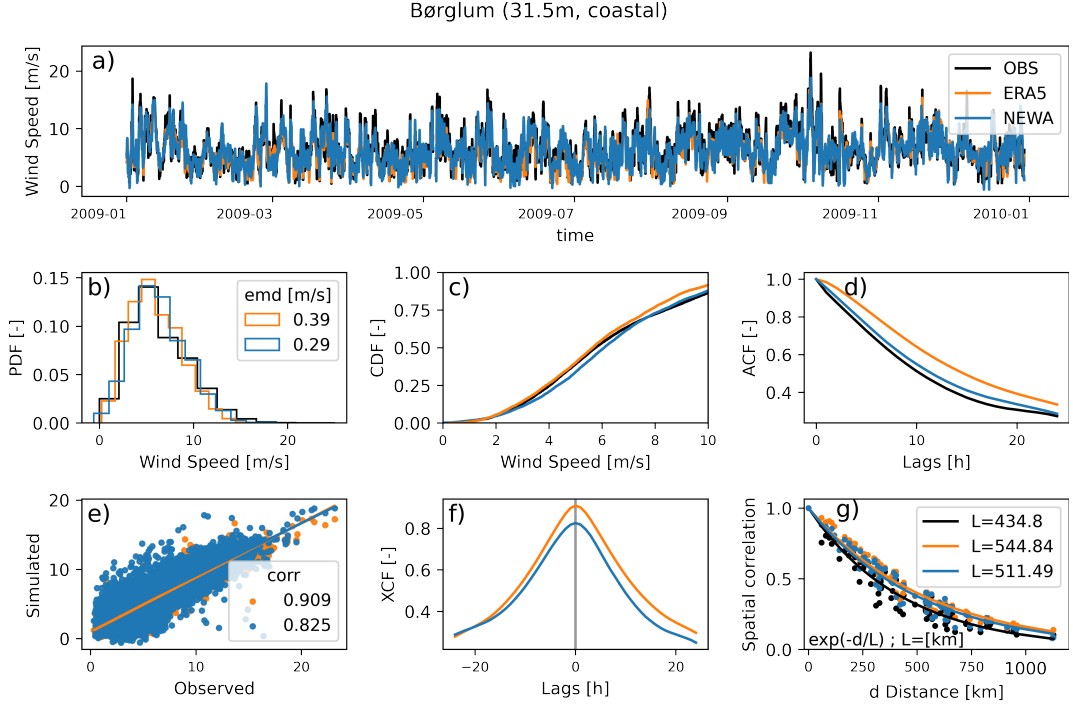

**Figure 2.** Example of the analysis done for one location (a-f) and all locations (g) comparing only observed data and two reanalyses (for simplicity) during one year. a) wind speed time series; b) probability density function (PDF) and EMD; c) cumulative density function (CDF); d) autocorrelation function (ACF); e) observed versus simulated wind speed; f) cross-correlation function; g) spatial correlations versus distance for all pairs of locations (dots) and their fitted curve (lines).

## 3 Results

This section presents the results for each metric described in Sect. 2.3. The tables contain the results for each data set (rows) and every site (columns). Columns 1 to 4 are located inland, 5 to 9 are located inland but close to the sea (hereafter named "coastal"), and 10 to 14 are offshore sites. The four last columns present the median over sites "onshore," "coastal," "offshore," and of "all" 14 sites, respectively. The color palette represents the best results in dark purple and the worst results in brown.





The rows are sorted by the column median "all," with the most accurate results on the top and least accurate results on the bottom of each table.

## 3.1   Correlation to measurements

In Fig. 3, time series from ERA5 reanalysis has higher correlation with measurements (median "all" = 0.91), followed by the "5km" experiment (0.89). The "3.3km" experiment is the least correlated among all data sets (0.80). The type of location

impacts the correlations. Sites offshore have higher correlations than coastal sites, which have higher correlation than onshore sites. The worst correlations for all simulations compared to observations are Sorø and Lindenberg, both met masts (43 m, and 98 m tall, respectively) located in forested sites. It reveals the difficulties of mesoscale models in simulating the effects of the forest on the flow dynamics, e.g., due to oversimplification and an unappropriated representation of roughness length, as it is discussed in Dellwik et al. (2014).

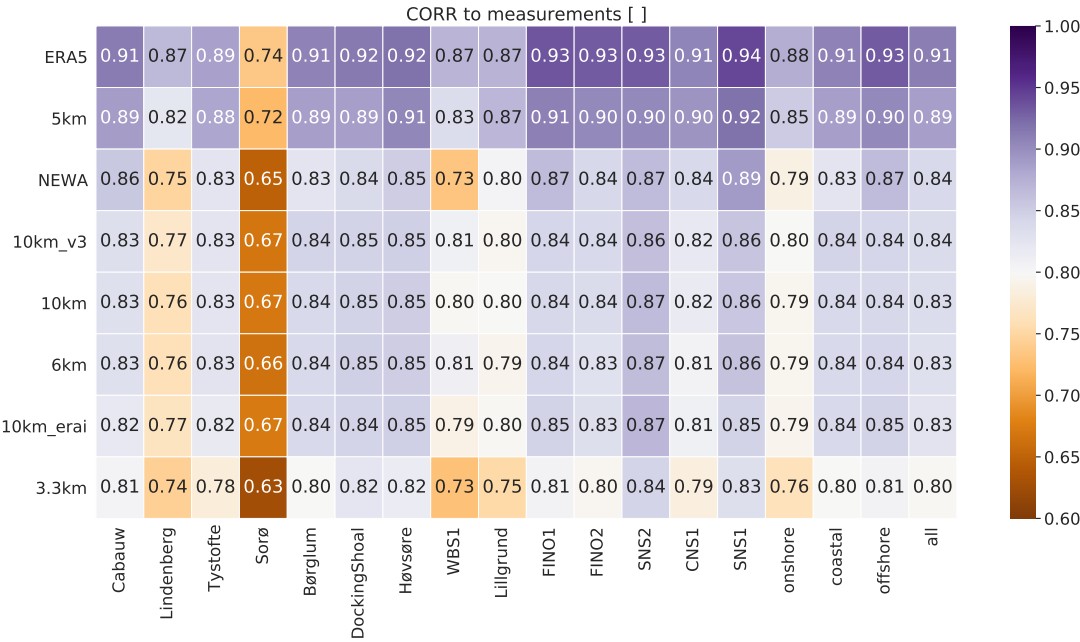

**Figure 3.** Correlation of simulated time series to measurements for the various experiments in Table 1 and reanalyses. The darkest purple colors are the most accurate metrics; the darkest brown, the least accurate metrics. The rows are sorted by the column median "all," with the most accurate results on the top and least accurate results on the bottom of the table.

## 3.2   Autocorrelation function

The ACF results (Fig. 4) show a clear spatial resolution impact. NEWA (3 km horizontal grid spacing) and "3.3km" experiment present the smaller errors in ACF ($\sim$ 0.031), while experiments with 10 km grid spacing ("10km_erai", "10km" and





"10km_v3") and ERA5 ($\sim 30$ km) present the larger errors $(0.047, 0.047, 0.049$ and $0.051$, respectively). As in the previous metrics, all simulations contain larger errors over forested sites (Sorø and Lindenberg). The ACF is simulated more accurately
in offshore sites, followed by coastal and onshore sites.

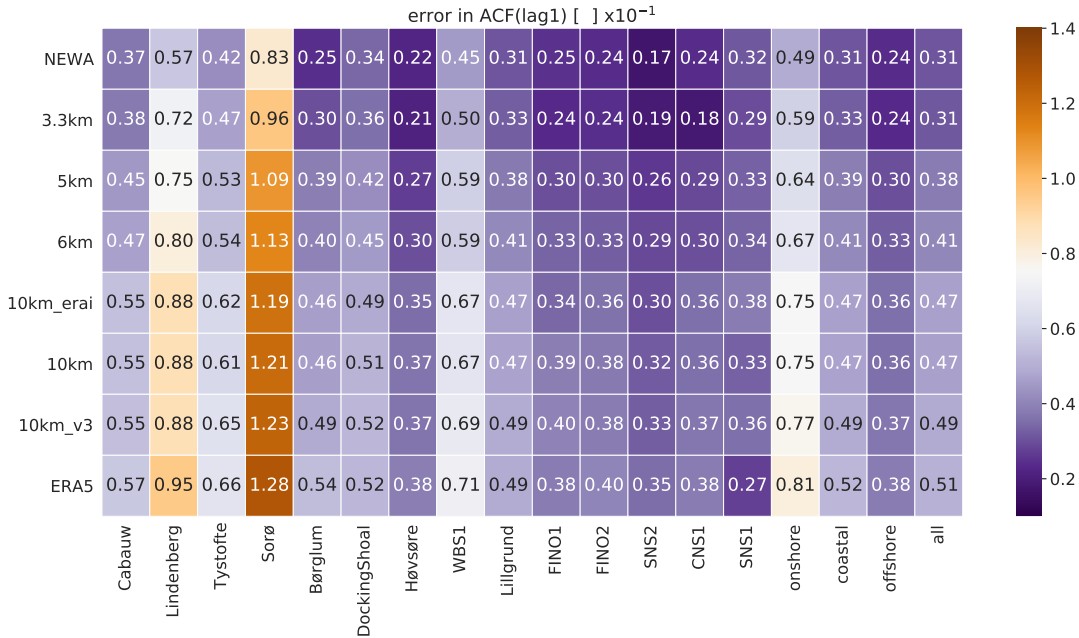

**Figure 4.** Error in autocorrelation function (ACF) at lag = 1 h of simulated time series to measurements for the various experiments in Table 1 and reanalyses. The darkest purple colors are the best results; the darkest brown, the worst. The rows are sorted by the column median "all," with the most accurate results on the top and least accurate results on the bottom of the table.

### 3.3 Standard deviation of first difference

As in the ACF analysis, the standard deviation (STD) of the first difference (Fig. 5) is impacted by the grid spacing of the simulations. The "3.3km" and NEWA simulations show smaller errors in this metric ($-0.30$ for both simulations), while experiments with 10km grid spacing ("10km_erai", "10km" and "10km_v3") and ERA5 time series present the larger errors
($-0.54, -0.55, -0.58$ and $-0.66$, respectively). All simulations underestimate the metric (negative values). Unlike all previous metrics, the STD of the first difference is more accurately represented over inland sites (especially for Sorø). The reason for larger errors in coastal and offshore sites can be due to the difficulties of mesoscale models in simulating turbulence over and close to the sea. On the other hand, the displacement height applied in the simulated time series over Sorø can mask the 1 h step changing errors. Originally, the time series interpolated at Sorø height (43 m) overestimates the wind speed above
the canopy of the trees at this forested site (Dellwik et al., 2014) and, therefore, produce errors in other metrics, such as correlations with measurements and wind speed distribution. We lowered the level of the interpolated time series using the





fixed displacement height of -20.5 m without taking into account that displacement height depends, among other things, on the wind speed (Dellwik et al., 2006). It is possible that the simulated time series exaggerates the turbulence at the displaced height and alleviates the underestimation in the STD of first difference metrics only for Sorø. For comparison, Lindenberg is a forest site, but we have not applied displacement height, and the errors are consistent with other inland sites. Even when the Sorø results are disregarded from the median "onshore", the onshore sites are more accurately simulated with respect to the standard deviation of the first difference than coastal and offshore sites.

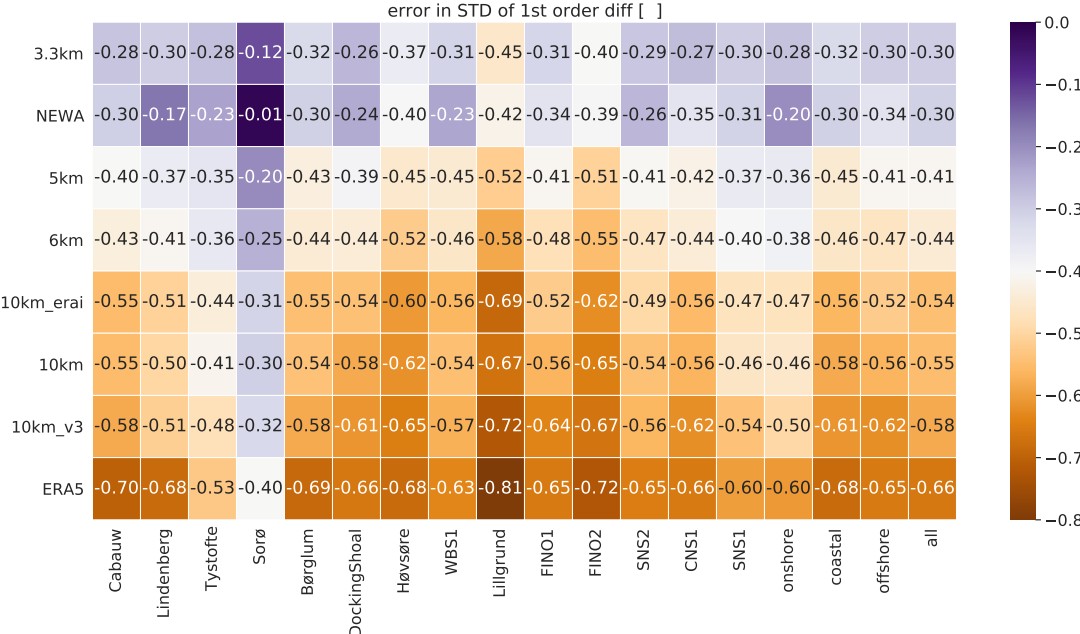

**Figure 5.** Error in the standard deviation of first difference (STD of 1st diff) between the simulated time series and the measurements for the various experiments in Table 1 and reanalyses. The darkest purple colors are the best results; the darkest brown, the worst. The rows are sorted by the column median "all," with the most accurate results on the top and least accurate results on the bottom of the table.

### 3.4 Wind speed distribution

The analysis of the wind speed distribution (Fig. 6) shows more homogeneous results over all experiments. The "3.3km" simulation has slightly smaller EMD ($0.27\,\mathrm{ms^{-1}}$), while its more equivalent simulation, the NEWA, presents an intermediate result ($0.29\,\mathrm{ms^{-1}}$). Larger EMDs are found for inland sites (especially Sorø and Tystofte); however, the coastal sites Docking Shoal and Lillgrund also present large EMD values. There is no clear sequence for the quality of wind speed distribution concerning the type of location, although onshore sites have larger EMD values in all data sets. All data sets underestimate (overestimate) the low (high) wind speed values (not shown) for inland observations in lower heights (less than $50\,\mathrm{m}$ tall).





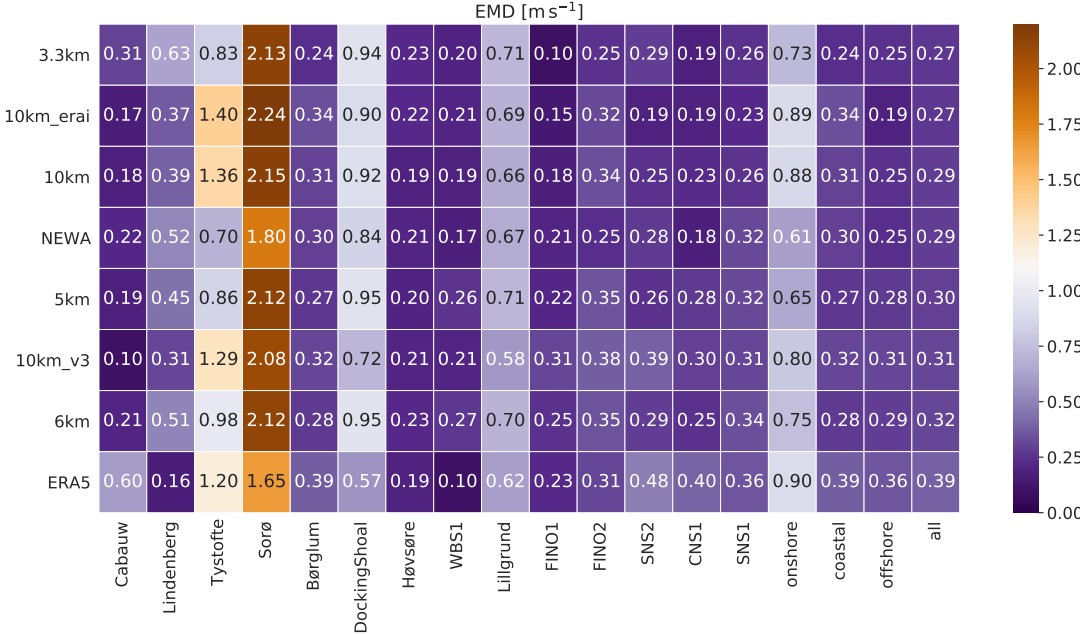

**Figure 6.** Earth mover's distance (EMD; $ms^{-1}$) between simulations and measurements for the various experiments in Table 1 and reanalyses. The darkest purple colors are the best results; the darkest brown, the worst. The rows are sorted by the column median "all," with the most accurate results on the top and least accurate results on the bottom of the table.

## 3.5 Spatial correlations


The metric to assess the spatial correlations (Fig. 7) is computed as described in Sect. 2.3. All simulations overestimate the correlations for most points, leading to larger parameters $L$, which agrees with previous studies (e.g., Murcia et al., 2022; Mehrens et al., 2016). The observed value of $L$ is 410 km, while the ones derived from the simulations vary between 496 km in the NEWA simulations and 541 km in the ERA5 time series. Mehrens et al. (2016) discuss the problem of the WRF model

being incapable of resolving wind variability sufficiently at higher frequencies due to the numerical smoothing, resulting in exaggerated correlations. Except for the NEWA time series, all simulations produce similar results despite horizontal grid spacing. The coefficient of determination r-squared ($r2$) for all fitted curves and the standard error ($e$) of the estimated parameters $L$ is also shown in Fig. 7.



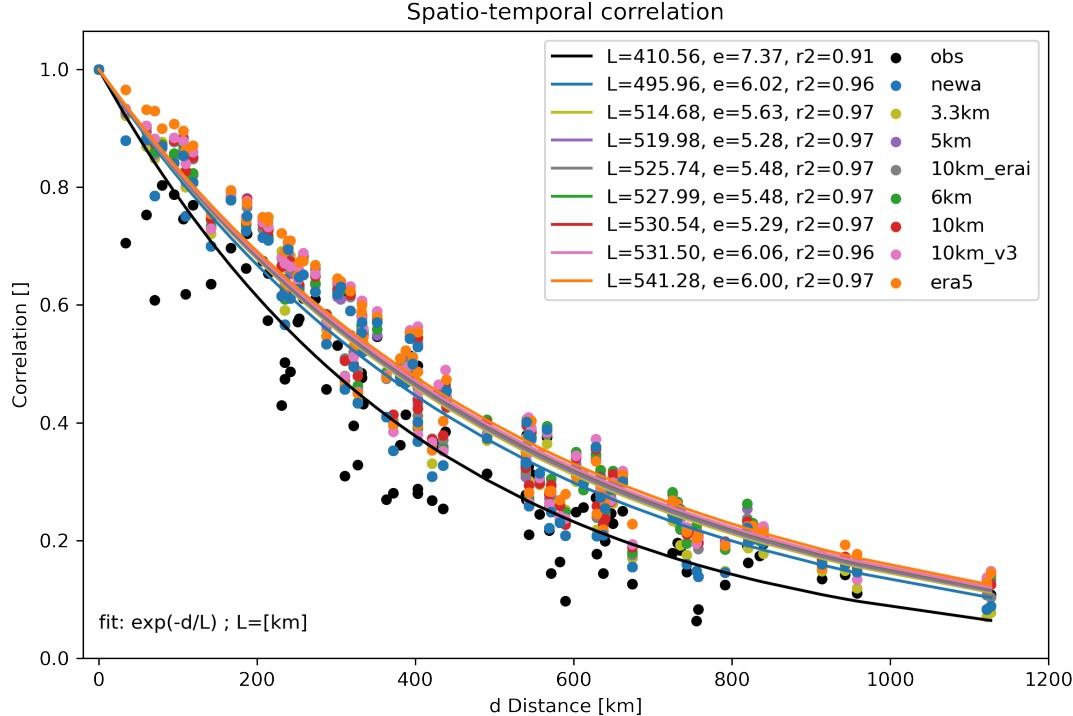

**Figure 7.** Correlation versus distance for each pair of sites (Table 2) and the fitted curves for the measurements (black) and the various model simulations in Table 1 and reanalyses. The estimated de-correlation length $L$ is also shown for each simulation.

## 4    Impacts from model setup

The boxplots in Fig. 8 represent the ranking among all simulated data sets in the first four presented metrics. The plots show the median (50th percentile), the first quartile (25th percentile), the third quartile (75th percentile), the maximum and minimum values, as well the outliers. The boxplots are ordered from best (lowest median, left) to worst (highest median, right) in all metrics.

The ranking for correlation to measurement (Fig. 8a) does not indicate a clear impact from spatial resolution. However,
ERA5 (coarser resolution) presents a higher correlation with measurements, which can be in part due to spatial smoothness. Also, the "ERA5" has the advantage of data assimilation, which periodically adjusts and approximates the simulation to the observations. Regarding the nesting arrangement, the "5km" experiment, which uses a nesting ratio of 1/3 (domain 1/domain 2), but setting domain 1 using 15 km, was more similar to the observed time series than its comparable "6km", which uses a ratio of 1/5 and resolution of domain 1 equal to 30 km (close to the reanalysis resolution). The "10km" experiment (also
ratio 1/3, but the resolution of domain 1 set to 30 km) correlates technically the same as the "6km". As the WRF model freely develops the simulations in the inner domains, it loses the correlation to the reanalysis. Double nesting amplifies this effect, such as the "3.3km" ratio 1/3/3). Using a smoother resolution jump, such as 15 to 5 km instead of 30 to 6 km, could be an



**Figure 8.** Boxplots of the metrics for all stations as a function of the model experiment: (a) correlations (CORR) to measurements; (b) error in the autocorrelation function (ACF); (c) error in the standard deviation (STD) of first difference and (d) Earth mover's distance (EMD). The model experiments are sorted as a function of their median, from the best to the worst.

advantage in keeping the correlations in the inner domain consistent with the driving reanalysis. Further tests are needed to confirm this behavior. Nevertheless, the ratio 1/3 – 15 to 5 km is double as computationally expensive as the 1/5 – 30 to 6 km or the 1/3 – 30 to 10 km. The comparison to the NEWA time series (1/3/3, 27 to 9 to 3 km) and the "3.3km" (1/3/3, 30 to 10






to 3.3 km) supports this hypothesis. However, the NEWA simulations are not directly comparable to the "3.3km" simulations because NEWA uses different choices of domains size and placement (e.g., domain 1 in NEWA is much larger than in "3.3km", and domain 3 is longitudinally longer in NEWA, while in "3.3km" is larger in latitude).

The boxplot of error in autocorrelation function (Fig. 8b) shows a clear impact from spatial grid spacing. The NEWA
and "3.3km" simulations present smaller errors than the measured time series. The 10 km grid spacing experiments and the "ERA5" time series show the most significant errors. Coarser resolution simulations exaggerate the correlations due to the inherent spatial smoothness of the atmospheric models, which can be seen in the results for all simulations (Fig. 4). The same interpretation can be made from the boxplot for the standard deviation of first difference (Fig. 8c), although in this metric experiments with very similar results, such as NEWA and "3.3km", and "10km" and "10km_erai" have inverted its ranking
positions.

The EMD boxplot (Fig. 8d) has the least conclusive ranking order among the metrics. There is no apparent influence from the spatial resolution in the wind speed distribution since the ranking alternates finer and coarser-resolution experiments. Both "5km" and "6km" present intermediate results and nearly identical values. From these results, there is also no significant indication of an impact on the simulated wind speed distribution, neither positive nor negative, from the complexity of nesting
(e.g., single versus two nested domains), the choice of nesting ratio, or the resolution jump.

The fitted spatial correlations (Fig. 7) show a clear distinction between the NEWA and the rest of the simulations. The NEWA time series presents the smaller value of parameter $L$ and the closest to the parameter determined for the measured points. A ranking of the simulations can be seen in Fig. 9, showing the NEWA simulations with the smallest ratio $L_X$ over observed $L_Y$, followed by all the other simulations with close overestimated results. As for the EMD, the spatial correlations explain
the ranking order, neither on the nesting choice nor the resultant spatial resolution. Part of the spread among the results from a single experiment comes from the use of various measurement heights. The results found for this metric agree with Murcia et al. (2022) that all simulations overestimate the spatial correlations and that the NEWA time series modeled this aspect of the time series more accurately than the ERA5 data set.

To check the consistency of the results in different periods, we recalculated all the metrics for the winter (Jan–Mar) and
summer (Jul–Sep) months. The results (not shown) keep a similar ranking order to the annual time series for all metrics, except for the EMD and spatial correlations. In any seasonal period considered, the EMD values range from approximately 0.3–0.4 ms$^{-1}$, but the ranking of the simulations is different (not shown). For the spatial correlations, Fig. 9 shows a different order for each considered period. Almost all simulations represent have higher correlations during winter months than summer months. For all simulations and the observed time series, the decorrelation length $L$ is larger during winter than during summer (not
shown). This could be explained by the larger spatial scale of winter versus summer atmospheric processes and their variability. Nevertheless, this result contradicts Solbrekke et al. (2020), although that study only includes correlations versus distances over the northern North Sea and the Norwegian Sea and a limited number of measurement sites.

All five WRF model experiments were repeated using the WRF model version 3.8, although only the "10km_v3" was included in the plots for comparison with "10km" (WRF version 4.2.1). The rank is unchanged from that with WRF V4.2.1
(with minor differences) for the correlations to measurements, error in ACF, and for the error in STD of first difference (not

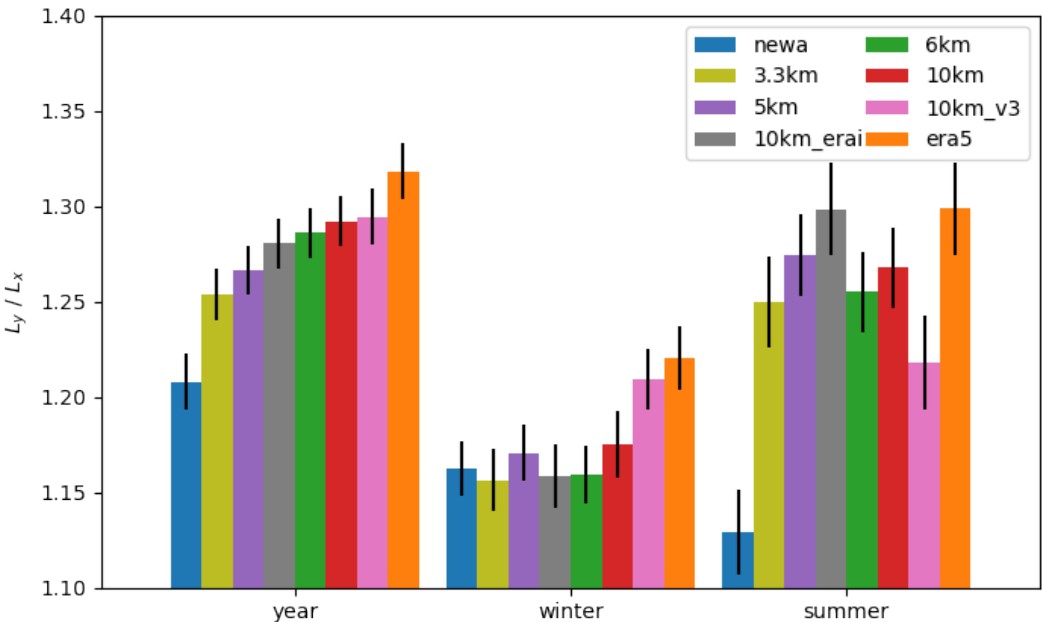

**Figure 9.** Ratio of simulated and observed characteristic length scales $L_Y/L_X$, both parameters fitted from curves in Fig. 7, and the standard error of the estimated parameters (bars). Winter refers to months Jan–Mar; Summer refers to Jul–Sep. $L > 1$ means that simulations overestimate the parameter $L$.

shown). However, the ranking order is changed for the EMD and the spatial correlations. Nevertheless, the conclusions for these two metrics do not show a clear impact from grid spacing or model nesting. The direct comparison between "10km" and "10km_v3" shows no clear effect from the two WRF model versions, and both experiments present a similar position in the rank for most metrics (Fig. 8).

Lastly, an experiment testing different forcing data ("10km_erai") was included to compare simulations forced by ERA5 versus ERA-Interim reanalyses. For all metrics, the "10km" and "10km_erai" present a close position in the ranking (Fig. 8). The three experiments with 10 km grid spacing are clustered among the ranks due to the same resultant grid spacing and model nesting.

## 5   Conclusions

To investigate how to improve the mesoscale modeling of wind time series over Northern Europe for power and energy system purposes, we performed a sensitivity study to various WRF model setups, including varying nesting configuration (1 or 2 inner





domains), nesting ratio (1/3 or 1/5) and resolution of the innermost domain (10, 6, 5 or 3.3 km). Simulations using different model versions and forced by different reanalyses are also explored. Five metrics relevant to wind power integration studies are presented for the time series derived from the WRF model simulations and compared to those from the New European Wind

Atlas and the ERA5 reanalysis. We also ranked the time series simulations metrics to identify significant factors controlling the simulation performance in their generated wind speed time series. Measured data from 14 sites over land, coastal and offshore locations in Northern Europe were used.

The value of the metrics at the considered sites shows increasing accuracy in onshore, coastal, and offshore locations considering all metrics, except for the standard deviation of the first difference. Sites located in forest landscapes and when mea-

surements are taken at lower heights generally have the more significant errors when compared to observations due to model deficiencies in simulating boundary layer processes near the ground in more complex terrain, in agreement with Hahmann et al. (2020b).

The evaluation of correlations to measurements indicates that strengthening the influence of the forcing from the reanalysis data on the mesoscale model simulation can be achieved by using a smooth transition between the computational domains.

Thus, a nest transition from 15 km to 5 km (domain 1/domain 2), is more effective than using 30 km to 6 km (considering the forcing data resolution close to 30 km) for maintaining the high correlations from the reanalysis in the inner domain. A comparison between simulations using three domains (30/10/3.3 km, and 27/9/3 km) confirms this result. However, the NEWA and "3.3km" simulations are not entirely comparable because they differ in the size of the outer domain. A large nudged outer domain will appear to be important for improving the correlation with observations in the inner domains. Still, our results do

not provide a systematic validation of this hypothesis. The ERA5-derived wind speed time series has the largest correlation to measurements for all sites, and this behavior is in agreement with Jourdier (2020). From experiences in weather forecasting, it is known that higher resolution does not always produce improved statistics (Mass et al., 2002) because the various metrics are sensitive to the smoothness of the time series.

The ranking order in the autocorrelation function and standard deviation of first difference is a function of decreasing spatial

grid spacing rather than the nesting arrangement. This is probably a consequence of the higher frequency of occurrence of convective processes in finer grid spacing domains, as it is discussed in Mass et al. (2002) and Vincent et al. (2013). For the wind speed distribution, the results are inconclusive for the impact from the model configuration or the spatial resolution on the quality of the time series. The analysis of the spatial correlations confirmed results from previous papers, that all simulations exaggerate the spatial correlations (Murcia et al., 2022; Mehrens et al., 2016) and that NEWA time series can simulate this

aspect more accurately than does the time series derived from the ERA5 reanalysis (Murcia et al., 2022). Mass et al. (2002) show that finer horizontal resolution leads to lower correlations due to a higher spatial variability. However, our results for spatial correlations do not find an explanation in the model setup and are sensitive to the period of the year. This could be because our tested grid spacings are very similar (from 10 km to 3.3 km) while in Mass et al. (2002) the simulation resolutions have a larger range from 36 km to 4 km.

As final conclusions, we found that the model configuration affects the value of the wind time series correlations with measurements metrics more than the grid spacing. Thus, we recommend ERA5 reanalysis over the mesoscale simulations for



studies where the correlations with measurements are essential. However, when producing mesoscale simulations for power and energy system purposes, a smoother resolution jump from outer to inner domains benefits the simulations by keeping it more correlated to the forcing reanalysis. This is especially relevant when the wind speed time series are combined with other series data (e.g., electric load or price time series). Finer spatial resolution simulations such as NEWA and "3.3km" may be best for applications where temporal variability has to be well modeled, such as power ramp analyses or voltage stability studies. For more accurate simulations in terms of wind speed distribution and spatial resolutions, NEWA presents more favorable results than ERA5.

Due to computational cost, many other details related to the model setup have not been tested. For example, we used the same size and position of the innermost domain for all simulations. Therefore, we did not test the sensitivity of the simulated time series to these aspects. Hahmann et al. (2020a) found that smaller domains in the WRF simulation tend to show smaller wind speed biases, but higher root mean square errors (RMSE) compared to observations. They claim, however, that it was unclear if this was resultant from the domain size or rather from the location of the boundaries in relation to the large-scale flow. Further tests including these two model setup aspects could point to improvements in modeling time series correlated with measurements since RMSE and correlations are related metrics. Additional numerical experiments on grid spacing could be carried out to clarify the potential impacts of horizontal resolution on the simulated spatial correlations and wind speed distribution. Finally, simulations using a much larger outer domain than the one in Fig. 1 and the same inner domain could explain the different performances between NEWA and "3.3km" (Fig. 9) in representing spatial correlations.

*Code availability.* The WRF model is an open-source code and can be obtained from the WRF Model User's Page. We used WRF versions 3.8 and 4.2.1 (Skamarock et al., 2008, 2019). The code modifications, namelists, and tables files we used are available from the NEWA GitHub repository (Hahmann et al., 2020a). The WRF model namelists and geofiles used in the simulations are temporarily available at DTU Data. When the paper is published, the content will be publicly available with a DOI. The code used in the calculation of EMD metric is available from https://pypi.org/project/pyemd/ (last access: 25 March 2022) (Pele and Werman, 2009).

*Author contributions.* GL: conceptualization, methodology, data curation, model simulations, writing (original draft, review, and editing). ANH: conceptualization, methodology, writing (review and editing), supervision. MJK: conceptualization, methodology, writing (review and editing), supervision, project administration.

*Competing interests.* The authors declare that they have no known competing interests that could influence the work reported in this paper.

*Acknowledgements.* The authors acknowledge support from the PSfuture project, Denmark (La Cour Fellowship, DTU Wind Energy). The authors would like to thank Oscar Manuel Garcia Santiago for supporting the coding of time series extraction from non-regular grids.



The following projects and organizations have kindly provided the tall mast data used for the evaluations. Cabauw data were supplied by the Cabauw Experimental Site for Atmospheric Research (Cesar), which KNMI maintains; FINO 1 and 2 were supplied by German Federal Maritime And Hydrographic Agency (BSH). Børglum, Høvsøre, Lillgrund, Sorø and Tystofte data were obtained from the Technical University of Denmark (DTU) database. CNS1 offshore data were provided by the NorseWind project (Hasager et al., 2013). Marine Data Exchange provided docking Shoal data maintained by The Crown State UK. Lindberg data were provided by the Tall Tower Dataset (Ramon et al.,

2020). The WRF model simulations were initialized using ERA5 and ERA-Interim reanalyses downloaded from ECWMF and Copernicus Climate Change Service Climate Data Store.



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
