# Peer review of "Evaluating the mesoscale spatio-temporal variability in simulated wind speed time series over Northern Europe"

_Wind Energy Science, 2022_

## Author Response (AR1)

**RESPONSE TO REVIEWER'S COMMENTS**

The authors thank the reviewers for their valuable comments, which have significantly improved the quality of the article. The author's comments below refer to the text in the submitted revised version.

**RC1:**

**General comment:**

This paper investigates how to improve the mesoscale modeling of wind time series over Northern Europe for power and energy system purposes by performing a series of WRF sensitivity studies. The results are interesting and certainly worth noticing by the wind energy communities. Therefore, I suggest minor revision before publication.

**Minor comments:**

Line 9: "high resolution" to "high temporal resolution"?

Authors: It refers to the spatial resolution, as in the previous sentence. The text in the abstract (L.9) was modified to clarify.

What is the simulation period? Need to include that in Section 2.1

Authors: The simulation period is one year (2009). Information added to the text in Sect. 2.1 (L.119).

What is the temporal resolution of the measured data (Section 2.2)

Authors: It was originally 10 minutes averaged temporal resolution and aggregated to hourly in the analyses (see L.136.)

**RC2:**

This is an interesting study which evaluates the ability of the Weather Research and Forecasting (WRF) model to accurately simulate near-surface wind speed timeseries. A range of model configurations (e.g. different nesting setups or boundary forcing) are compared to global reanalysis data using a number of statistics relevant to someone who might want to simulate timeseries of wind power. The paper is well written and provides clear and useful results. The experiment setup is very well thought out and I liked the choice of metrics to validate the models against. I have a few minor comments below for the authors to consider before publication. The main theme of my comments is that given energy-meteorology is a very interdisciplinary field, a bit more information and clarification of some of the modelling steps may increase the readability and uptake of the research.

**Minor comments:**

L32: Can you comment on how 'extra information can be added by combining microscale with mesoscale data' ?

Authors: The effects of a more detailed terrain in the wind speed distribution, such as blockage effect and speed-ups in mountain regions, can be added to the mesoscale data by applying a corrective factor based on microscale data (e.g., using gridded mean wind speed data). The text in Sect. 1 (L.35-37) was modified to clarify.

L34: Can you be more specific on the definition of short-term in this context?

Authors:  It refers to minutes to seconds-scale variability. Information added in Sect. 1 (L. 39-40).

L35-45: Can you comment on the datasets which performed best from these studies?

Authors: Both studies found that ERA5 is well skilled and the best correlated but presents deficiencies in simulating wind speed in areas with more complex terrain if not corrected. In general, AROME and COSMO-REA6 are ranked best in the metrics analyzed in Jourdier (2020). NEWA is better at representing the spatial correlations and the temporal properties of the wind speed time series at individual locations or WPPs in Murcia et al. (2022). This information was added to the text in L.51-55.

L46: This is a long sentence which is hard to understand in its current form, consider breaking into two?

Authors: The text was modified as suggested in Sect. 1 (L.56-60).

L55: [The results] – is this in reference to all three of the previously mentioned papers?

Authors: It refers to Koivisto et al. (2021) (L.66). The text in L.67 was modified to clarify.

L63: This work [focuses] on …

Authors: The text was modified as suggested in L.71 and L.75.

L70: It is probably worth adding in a comment that your study region is Europe in this paragraph.

Authors: Comment added to the text in L.78.

L78: Can you highlight the key updates between the two WRF versions used that might influence the results here?

Authors: The WRF version 4.2.1 was the latest version available when the experiments were run, and it contains modifications in the model physics (including the MYNN scheme), that could lead to differences in the results when compared to the version used in NEWA production (v. 3.8). For this reason, the experiments were repeated using version 3.8, which made it possible to check for impacts from the model version and a controlled comparison with NEWA dataset. This information was added to the text (L.88-96)

L79: A couple more details would be useful, e.g. the frequency of the nudging. This is potentially of relevance when thinking about the correlations.

Authors: The data used for the nudging has a frequency of 6 hours. The information was added to the text (L.97-98).

L80: A quick summary of the WRF validation would be useful to confirm that the model captures the right phenomena for the right reasons, and any previous comments on things of relevance WRF may struggle with.

Authors: A brief comment was added to the text (L.99-101).

L99: Can you mention the model levels that WRF is outputting data and what you have from ERA5? This would give a sense of the importance of the logarithmic extrapolation. Also does the logarithmic extrapolation make assumptions about atmospheric stability? If so can you comment on this, or if it might be more appropriate at certain times of the year?

Authors: WRF simulations and ERA5 and NEWA reanalyses use the two closest levels for the vertical logarithmic interpolation. The NEWA data set and the WRF simulations have outputs in several fixed levels ranging from 25 m to 250 m. From the ERA5 data set, only two fixed height levels (10 m and 100 m) were used and an extrapolation is assumed for sites taller than 100 m. No assumption on the atmospheric condition is used, and the same process is applied every time step throughout the year. This information was added to the text (L.123-127).

L112 and in L303: Is there anything remarkable about the year 2009 that may influence your results? I appreciate it isn't possible to do multi-year simulations due to computational requirements and lack of validation data, but this does give you a very limited sample. Some comments on your chosen year (notable extreme events, or lack of these) would be useful for context.

Authors: Extra information was added to the text in Sect. 2.1 and Sect. 5 (L.118-120 and L.345-346, respectively).

Section 2.3: I like the metrics. Can you comment somewhere that because of shapes of power curves errors at certain parts of the distributions matter more than others? For example errors below the cut-in speed are less important than in turbine ramping regions.

Authors: A comment was added as suggested in Sect. 2.3 (L.160-163).

L144-155 are a bit repetitive of the metric descriptions so these could be combined.

Authors: Unnecessary information was removed from Sect. 2.3 (150-171). The text in L.177-183 was modified accordingly.

Fig 2: Define EMD in the caption.

Authors: Definition of EMD included in the caption of Fig. 2.

L 163: either [have higher] or [has highest] might be better?

Authors: Text modified as suggested in L.195.

L166: 'It reveals the difficulties of mesoscale models in simulating the effects of the forest on the flow dynamics,' – but surely ERA5 isn't doing a very good job of this? Can you clarify any other reasons for why ERA5 is highest?

Authors: The reasons why ERA5 correlates better with measurements, such as spatial smoothness and data assimilation, are briefly discussed in Sec. 4 (L.247-250). For all simulated sites, ERA5 has higher correlations than the mesoscale simulations. However, the reanalysis shows a similar deficiency in simulating the correlations at Lindenberg and Sorø (see Fig. 3).

L175: Are the autocorrelations a lot worse than the correlations for ERA5 due to issues in simulating the variance of the data? It might help to show a short timeseries of the data to illustrate some of these issues?

Authors: The results show that the autocorrelations are exaggerated in coarser-resolution simulations due to the spatial smoothness. This is briefly discussed in L.265-266.

L180: 'The reason for larger errors in coastal and offshore sites can be due to the difficulties of mesoscale models in simulating turbulence over and close to the sea' – This could be a reason, but you haven't explicitly tested for this. Is there a reference you can put to support this comment?

Authors: A reference was added to the text in L.215.

L199: I think this not-shown result is of high relevance. It would be interesting to break these results down into different regions of a power curve e.g. before cut-in, ramping, rated, and above cut speeds out to see if different datasets have larger errors in different regions. As this may be useful in deciding a particularly model setup. This is most relevant for the EMD metric.

Authors: A figure showing the wind speed distribution and EMD metrics for two inland sites at lower heights (Fig. 7) was added to the manuscript.

L216: 'Also, the "ERA5" has the advantage of data assimilation, which periodically adjusts and approximates the simulation to the Observations' – but is this information not then passed through to the WRF simulations? Can you clarify?

Authors: The data from observations is indirectly passed thru the model domains. But the nudging for wind is only carried out above the boundary layer. In addition, the comparison between the simulations forced by ERA5 versus Era-Interim has shown that the results from the inner domain are dominated by the WRF model solution rather than the reanalysis, and both experiments ranked equally in almost all metrics (See L. 299-300). However, the results suggest that the resolution jump from outer to inner domains impact how the correlations to measurements are passed through the domains (See L.250-260).

L278: Can you consider rephrasing this sentence as it's currently a bit confusing.

Authors: The text was modified in L.320-324.

L289: 'appears to be' rather than ' will appear to be'

Authors: The text was modified as suggested in L.332.

L305-315 These are very nice key points. Can you maybe bring these headline results to the start of the conclusions and then unpack them afterwards? This is the key information the reader will come to this section looking for first.

Authors: The paragraph was moved to the beginning of the Conclusions (L.312-319)

---

## Author Response (AR2)

**RESPONSE TO ASSOCIATE EDITOR'S COMMENTS**

The authors thank the associate editor for the valuable comments. The comments below refer to the text in the submitted revised version.

First, some defense of the choice of 2009 (aside from the availability of observations) would be helpful. The revised text simply says (analysis not shown) - please provide some support, even if in an appendix or a reference to another published study.
Authors: A figure (Fig. A1) was added to the Appendix (p. 19), and the text was modified in Section 2.1 (p. 4) to clarify that the selected year is a typical one in terms of wind speed over the region studied and should not affect the results.

Second, the conclusions should repeat the point that some parts of the wind speed distribution matter more than others (because of the variability in the power curve) and suggest future work with appropriate weighting.

Authors: The suggested comment was added to the Conclusions (p. 18). "Finally, because of the shape of the power curves, a further analysis focusing on the errors (i.e., EMD) on certain parts of the wind speed distribution that contribute the most to energy production could be carried out by assigning higher weights to values between the cut-in and cut-out wind speeds"

Finally, please prepare a DOI or github with the WRF namelists and supporting data as suggested in the Data Availability section.

Authors: The reference with DOI was added to the Code Availability section (p. 19).